# A Qualitative Evaluation of the Implementation of an Oral Care Program in Home Care Nursing

**DOI:** 10.3390/ijerph20032124

**Published:** 2023-01-24

**Authors:** Lina F. Weening-Verbree, Annemarie A. Schuller, Sytse U. Zuidema, Johannes S. M. Hobbelen

**Affiliations:** 1Research Group Healthy Ageing, Allied Health Care and Nursing, Hanze University of Applied Sciences, 9714 CA Groningen, The Netherlands; 2FAITH Research, Groningen/Friesland, Hanze University of Applied Sciences, 9714 CA Groningen, The Netherlands; 3Center for Dentistry and Oral Hygiene, University Medical Center Groningen, 9713 AV Groningen, The Netherlands; 4TNO the Netherlands Organization for Applied Scientific Research, 2333 BE Leiden, The Netherlands; 5Department of General Practice and Elderly Care Medicine, University Medical Center Groningen, University of Groningen, 9700 AD Groningen, The Netherlands

**Keywords:** dental hygienist, experienced impact, home care nursing, implementation of oral care, older people, oral care program, personalized oral care, qualitative research

## Abstract

An Oral Care Program (OCP) was implemented in home care nursing teams in a northern province of the Netherlands to improve the oral health and hygiene of older people who make use of formal home care in 2018–2019. The aim of the current study was to evaluate the experiences of the stakeholders involved (older people, home-care nurses and dental hygienists) and to report the experienced impact of OCP, with a qualitative approach. Three dental hygienists, nine home care nurses, and eight older people were interviewed with semi-structured interviews, which were audio recorded, transcribed and analyzed using thematic analysis. The codes derived were grouped into nine main themes. OCP was experienced as mostly positive by all stakeholders involved. The educational part lead to more awareness towards oral care, but should be repeated regularly. Personalized oral care plans for older people were experienced positively, however, obtaining oral care behavior changes appeared to be difficult. Collaboration between dental hygienists and home care nurses lead to a positive experience from both sides The method and intensity of collaboration varied between the teams. To provide better access to oral health care for older people in the community, a long term collaboration between home care nursing teams and dental care professionals in their working area should be established.

## 1. Introduction

Multiple associations between oral and general health have been reported in the literature; in older adults these associations are more obvious [1,2,3,4,5,6]. Oral health care in older adults is therefore of importance. Oral health seems to deteriorate with increasing age, and older people tend to visit dental professionals less frequently than they did when they were younger [7,8,9,10,11] For frail older people, these problems are even more pronounced [8,12,13]. The healthcare system in the Netherlands is regulated by the Care Insurance Act, which ensures that every Dutch citizen gets access to basic health care. Dental health care for adults (18 years and over) is, however, not covered. Adults, including community-dwelling older people, have to pay their dental care costs personally or they can make use of an additional dental care insurance to cover (some of) these costs [14]. When people depend on formal home care, the same rules still apply regarding the insurance of dental treatment. However, according to the to the home care nursing guidelines, (support with) daily oral care is part of home care nursing [15].

In order to improve the oral health of frail community dwelling older people, we implemented an Oral Care Program (OCP) in home care nursing teams in a province in the northern part of the Netherlands [16]. The OCP targeted home care nurses and older people using home care nursing, and was based on current oral care guidelines in home care nursing [15]. Three groups of stakeholders were targeted: (1) older people: raising their awareness about oral health with a magazine and a weekly calendar, both with information about oral health care, and providing a personal oral care plan based on an oral screening by a dental hygienist. The OCP provided dental screening for the older people free of charge. In addition, older people were supported with instruction cards and oral care products including toothbrushes and denture brushes; (2) home care nurses: educational sessions and practical instructions about oral care were provided to the home care nurses in order to increase knowledge, attitude and skills; and (3) dental hygienists, collaborating in care with home care nurses. A dental hygienist was designated to each of the home care nursing teams; home care nurses could question and discuss their experiences with oral care with their older clients with the dental hygienist of their team without costs.

The results of the quantitative evaluation of the implementation of the OCP has been described before [16], and showed a knowledge increase in home care nurses as well as improvements in the oral health of the participants. To effectively implement a program such as OCP in the future, it is of significant importance to gain detailed insights into the opinion of the stakeholders concerning the program [16]. From earlier qualitative studies, although performed in long-term care facilities, the context of implementation, the experiences with the oral care programs implemented, and the evaluation of the practicability of the oral care programs provided valuable insights with regard to implementations, such as the requirement of ongoing education and policy implications [17,18,19]. These qualitative evaluations show that, despite a positive attitude, difficulties exist in performing daily oral care and organizing oral care [17,18,19,20]. Currently, as far as we know, no report of any qualitative evaluation of oral care programs implemented in home care nursing is available. A better understanding of the implementation of OCP will contribute to achieving a more sustainable practice, improve future implementation of OCP, and contribute to the better integration of oral care in home care nursing.

Therefore, the aim of this study was to evaluate the experiences of the stakeholders involved (older people, home-care nurses and dental hygienists) with the implementation of the Oral Care Program and to report on the impact of the Oral Care Program, using a qualitative approach.

## 2. Materials and Methods

### 2.1. Study Design

A qualitative study using thematic analysis [21] of semi-structured interviews was conducted after the quantitative measurements were completed (October 2019–February 2020). The study was performed in six home care organizations in a northern part of the Netherlands (Friesland), and 21 home care nursing teams participated in OCP with their home care nursing team members (approximately 260 home care nurses). Approximately 1200 older people were reached by the OCP, of which 190 participated in the quantitative study. The implementation and follow-up measurements per home care nursing team took at least six months to complete. The primary participating home care nursing teams started with the OCP in January 2018, with the final implementation of the OCP starting in February 2019, finishing by September 2019. In Figure 1 we have summarized the interventions of the OCP used in the quantitative study [16] in order to frame the participants’ experiences reported in this paper. The quantitative results of this study are reported elsewhere [16].

### 2.2. Participants

Members from all three stakeholder groups (dental hygienists, older people and home care nurses) who were involved in the implementation of the Oral Care Program [14] were selected by two independent external researchers to participate in a semi-structured interview. 

Older people and home care nurses were selected for the semi-structured interviews using a combination of both purposive and convenience sampling, but from multiple teams, in order to represent a wide range of home care nursing teams; however, within the teams, the nurses with the most experience with OCP were selected. The interviewers contacted the home care nursing teams intentionally to ensure a wide range of teams in the area where OCP was implemented, and then asked them if they might be willing to participate in the interviews (convenience).

Three of the seven participating dental hygienists who had been closely involved in the implementation of the Oral Care Program were approached for the semi-structured interviews as well.

The interviewers had no relationship or conflict of interest with the interviewees. Both interviewers were female, possessed Master’s degrees and were experienced interviewers. 

### 2.3. Data Collection

Semi-structured interviews were conducted at the older peoples’ homes or the workplace of the dental hygienists and of the home care nurses [16]. For all three groups, different interview guides (in Dutch, available upon request from the first author) were prepared prior to the interviews. The interview topics and questions were related to the implementation of OCP and were drafted during brainstorming and discussion sessions within our research group, together with the interviewers. Topics included the reasons for or motivation to participate in OCP, experiences with OCP and its interventions, communication and collaboration between the participants/stakeholders, the implementation of OCP in general, the contribution of OCP according to the participants, and the future potential of OCP. 

The interviews were audio recorded and lasted no more than one hour. To report the study, we used the COREQ (COnsolidated criteria for REporting Qualitative research) checklist, Appendix A [22].

### 2.4. Ethics

Written informed consent was obtained before the audio recording, and all participants were also verbally informed before the audio recording that they could cease the interview at any time, and that they should feel free to express their opinions and experiences. All data were processed anonymously, and privacy was respected according to the requirements of the Dutch Personal Data Protection Act. Approval was given by the Medical Ethical Committee of the University Medical Center Groningen for this study (study number 201700693). To protect the identity of the interviewees, we have not reported any background data, such as age, home care nursing team or working area. However, we did ask the dental status of the older people interviewed and in accordance with the quantitative study, the majority of older people interviewed had dentures and only one participant had their natural teeth at the start of OCP; during the project, a partial denture was made, and two older people were having dentures and natural teeth/dentures on implants.

### 2.5. Data Analysis

After the audio recording, all interviews were transcribed and anonymized. The data were analyzed according to thematic analysis [21] using Atlas.ti 9 (ATLAS.ti GmbH; Berlin, Germany). Firstly, all data were coded and processed by two independent, qualified and experienced researchers who conducted the interviews. The first coders used Microsoft Excel (Microsoft Corp., WA, USA) not Atlas.ti 9, and made a visual presentation of the findings to the participants involved in the implementation of OCP. Secondly, the first author of this paper coded and analyzed the data in Atlas Ti 9 using the codes and visualization of the first analysis, in addition to other codes. From the codes, themes per stakeholder group (dental hygienists, home care nurses and older people) were derived. Thereafter, data were combined in order to assign core themes. This process ended when data saturation was reached and no more themes were added. Finally, quotations were added to the main reported themes. Differences in interpretation were solved by discussions amongst all authors involved in this paper.

To highlight major thematic codes, they are in bold text. If a code was present in all three stakeholder groups, the text is in italics.

## 3. Results

Interviews with three dental hygienists, nine home care nurses and eight older people were conducted, and saturation of the data was reached. No new codes or code groups were added after the analysis of the first 12 interviews, which consisted of data from the interviews of seven older people, two dental hygienists and three home care nurses. The codes we identified were grouped into nine themes (Table 1). Three sections could be distinguished in the themes: the participation of stakeholders involved, the experiences with and results of the OCP, and a section about the follow-up of OCP.

### 3.1. Participation in OCP

The home care nurses mentioned that a small proportion of clients had participated in the OCP, but that the older people who participated were positive and enthusiastic about the OCP and the home visit of the dental hygienists. Older people had been encouraged to participate by the home care nurses since the nurses themselves were motivated to improve the oral health of their clients. It was mentioned that it was unfortunate that OCP was age limited (70 years and older), since home care nurses and dental hygienists would have liked to offer the program to all people, irrespectively of age. In addition, it was mentioned that if a younger population could be reached, prevention would possibly be more useful and more timely. The cost for clients was mentioned by almost all interviewed home care nurses, sometimes as a barrier (regular dental care is too costly or older people are afraid of costs of OCP, despite the OCP being free of charge) or sometimes as a facilitator in the OCP, promoting that the dental screening in OCP was free of charge. The costs for teams were mentioned by six of nine home care nurses (education takes some time with the team providing care), but most home care nurses were positive; for instance, with regard to the notion that OCP should be offered to home care nursing teams in general by health insurance companies. One other home care nurse thought that the expenses related to OCP were a waste of money and time.

Older people who participated in the OCP were involved for several reasons; some were curious about the possibilities of oral care or because they already experienced oral problems. Interviewer: *‘…and for you, what were the reasons to participate in the Oral Care Program?’* Older person 8: *‘well, because, I thought, maybe there is something that I can do about it (an oral health problem).’* Interviewer: *‘if I am correct, the home care nurses asked you to participate, why did you decide to participate?’.* Older person 2 *‘well, you can always learn something’.* Some older people were mainly interested because of the social aspect of the visit of a dental hygienist. Interviewer: *‘and what do you want?*’ Older person 3 *‘well, just that she visits me again’.* Interviewer: *‘another visit from the dental hygienist? Is that for your oral health or because of the social aspect?’* Older person 3: *‘because of the social aspect’*.

The mean age of the older people participating in the project was over 80 years, and some were (very) frail. One of the dental hygienists mentioned *‘nowadays you only are referred to a nursing home when your health is really bad…so to speak…you are completely dependent on other people. It was so good to visit older people of 80, 90 years old…they have physical problems…but still they live in the community, with support of home care nursing. It was an enjoyable way of meeting older people in their home situation and be able to support them with oral care’* (Dental Hygienist 1).

Two of the three dental hygienists interviewed mentioned the financial rewards for the dental home visit of the older people; the making of appointments and personalized oral care plans was not in proportion; and they stated that higher financial rewards would have been more appropriate to them.

### 3.2. Experiences with Interventions and Impact of OCP

Some home care nurses mentioned that they did not see any products or care plans themselves, because of inaccurate information transfers of the personalized oral care plans from the dental hygienists to the home care nurses or because older people did oral care themselves. The home care nurses stated that the time investment in the OCP for them was mainly due to the documentation for the study. Also, one of the home care nurses mentioned that participating in the OCP was not that time consuming at all, because as a team they had good contact with the dental hygienist.

However, not all OCP interventions were received well, as some older people and home care nurses mentioned that older people liked to receive the oral care products and instruction cards, but the magazine and calendar fonts and colors were difficult to read for older people. The background colors and font colors were too similar. Personalized oral care products (e.g., denture cases for denture wearers) and oral care instructions were highly valued by all participants of the interviews; however, some found the denture case difficult to open. The hourglass and calendar were less appreciated and the interviewees considered these mainly as ‘useless’. The instruction cards were not understood by some older people; as they did not recognize their oral situation in the generalized images on the cards.

Dental hygienists considered scheduling the home visits, undertaking the home visits, and the time investment, as ‘intense’. Dental hygienists indicated that screening the oral health of older people to complete the personal oral care plan was useful, yet they believed a category for implants and details about dentures would be beneficial.

### 3.3. Oral Health (of Community-Dwelling Older People)

All stakeholders mentioned that when older people have full dentures or when they have no complaints, they tend not to visit a dental professional on a regular basis. According to the dental hygienists, some of the older people were experiencing oral problems and some dentures were more than 40 years old, but this was not seen as reasons to visit a dental professional. Dental hygienist 2: ‘Some people, for example, when they had toothache, I tried to motivate them to visit a dental professional. But…I saw people with really, really neglected oral health. That I thought… this is the rock-bottom of my career, that denture looked so awful dirty.’

Dental hygienist 2 shared her view on what she had experienced while providing the oral OCP screening: ‘the demand for oral care amongst older people and home care nurses is missing, while we as dental care professionals, see that the need for dental care at home is a reality’. Dental hygienists mentioned that they had expected to see more people with natural teeth. They hypothesized that maybe some poor older people were probably ashamed of their dental status and therefore decided not to participate in OCP. However, this was not substantiated by the information retrieved from the older people themselves.

All interviewed stakeholders mentioned that oral health (care) is of minor importance or it is just not possible to maintain oral care at a certain level when general health is compromised; cognitive impairment, respiratory problems and mobility problems were also mentioned by participants.

### 3.4. Oral Care and Oral Care Behavior

Oral care behavior change in older people appears difficult according to all the interviewees. Interviewer: ‘and did you apply the advice from the dental hygienist? If I am correct, she told you about that?’. Older person 2: ’yes, we received an instruction card on how to brush. But let’s be honest, you brush for many years and you keep brushing that way’.

The home care nurses are quite clear about their involvement in supporting oral care (home care nurse 1): ‘autonomy and directing yourself are important in the Netherlands and stimulation to care for themselves. That means, that if an older person can perform oral care by himself, we are not taking over…enough older people do oral care for 30 years their own way and they say…well, I am not going to change that…but reactions on OCP and advice were mainly positive. That OCP was free of charge was very helpful; costs always play a role’. One of the home care nurses (nurse 4) who worked as a team leader and also provided home care nursing suggested that the demand and need for care in general, should also be addressed by the older persons themselves: ‘some are enthusiastic, some are reserved. Some (home care nurses) say, well, yes, that is what older people themselves should do, because they live independently in the community’.

The dental hygienists mentioned that older people tend to have their own habits and to change that behavior is difficult. Interviewer: *‘and what was the main reason that they (older people) did not follow your advice, or what did you hear about that?’* Dental hygienist 3: *‘that it is unnecessary. And (the older people said) I have never had anything done about it. And it doesn’t hurt me. And yes, I (dental hygienist) think also not knowing how that works and not feeling up to it’*.

### 3.5. Knowledge and Education

Home care nurses appreciated the educational sessions, and to them, it increased their knowledge and awareness of oral care. Also, home care nurses indicated that due to this education, they are more alert to oral problems in older people they care for. A well-known educational tool for improving oral care is training to brush your colleagues’ teeth, yet the home care nurses indicated that brushing your colleagues’ teeth can be awkward and uncomfortable. They suggested that practicing oral care on manikin heads was probably a better idea. The home care nurses thought that when OCP data collection is finished and no educational sessions are planned, their commitment to OCP or oral care will probably decrease. Home care nurse 2: *‘well, it made the team extra aware of oral care and we received new tips and tricks, that they can also pass on to older people’.*

One of the dental hygienists (dental hygienist 2) made the following comment about the education: ’well, that was really nice. People were quite enthusiastic and surprised what we can add with oral health care. That was really nice, that you know, hey this is new to them (the home care nurses), so that they can apply that in the daily nursing care’.

### 3.6. Awareness (of the Importance of Oral Care)

Awareness was used as code quite frequently by multiple participants and therefore identified as a separate theme in itself. Home care nurses and older people thought that the OCP increased awareness of oral care and the importance of oral care. The home care nurses mentioned that awareness increased due to the educational sessions, but this could be a temporary effect over the time the OCP was in progress. Home care nurse 7: *‘at least it (oral care) has been brought to our attention and that made a number of people aware of its importance. Also, older people became more aware of the importance of oral care;* Older participant 8: *‘Yes, more aware, yes, indeed. I have a new dental hygienist, a woman. She visits me now.’*

### 3.7. Communication and Collaboration

The OCP encouraged professional discussion on oral health and oral care amongst home care nurses, which previously was not common. Before the OCP, none of the interviewed home care nurses were in collaboration with general dental practices or dental hygienists. They all appreciated that the OCP connected them with dental hygienists. The collaboration of home care nurses with general medical practitioners was mentioned to be common, but not when it comes to dental problems. However, communication between the dental hygienists and home care nurses differed greatly among the home nurse teams. Some home care nurses and dental hygienists were closely in contact and informed each other during the OCP, whilst other nurses were unaware of the OCP dental hygienists’ activities and advice given to the older people. Dental hygienist 2: *‘There was no contact with the home care nurses, but I can also blame myself for that, I have a very busy schedule and I did not initiate the contact’.*

### 3.8. Access and Barriers to Dental Care

The OCP provided an oral screening for older people that was much appreciated by those who participated in the interviews. Older people mentioned that they tend to seek dental care only when they have oral problems; when they have no complaints, they do not visit a dental care professional or they have no idea where to go for dental care. Older participant 5: *‘No. I don’t know exactly where to go for that oral infection, to the hospital or the general practitioner, I just don’t know. I am willing to go to the dental…the person who repairs my prosthesis, that it is solid in my mouth again. But eh, for the infection….well I don’t know*’.

The OCP was received positively, as it informed about the possibilities of oral care at home (mobile oral care), home visits of dental hygienists, and about the need for regular screening. Older people with dentures mentioned that they visit a prosthetist directly when they have a problem with their dentures. Home care nurses highlighted the financial barrier for older people in their teams; in some areas older people are poor and dental care is a ‘luxury’ they cannot afford.

### 3.9. Follow-Up of OCP

In this theme the follow-up suggestions of the participants were gathered in one overarching theme of ‘follow-up of OCP’. According to the dental and medical situation, personalized oral care and the number of visits should be included in a follow-up Oral Care Program, as mentioned by dental hygienists and home care nurses. Educational sessions are needed to maintain motivation and awareness for oral care in the home care nursing teams. Some home care nurses think that this could also be in e-learning programs, and yearly scheduled education was suggested. Home care nurse 5: *‘Yes, you are more aware, for example, with an intake of a new client, now we have added a standard question ‘who is responsible for your oral care and who does oral care?’.*

In the OCP, dental hygienists with experience in geriatric dentistry were recruited, but the nursing home teams mentioned that long-term collaboration should be established with dental care professionals in their working area.

## 4. Discussion

### 4.1. Summary of Findings

From the semi-structured interviews with all groups of people involved in the OCP, it can be concluded that the OCP in itself was experienced positively, and the experienced impact was positive in the home nursing teams and among the older people and dental hygienists. Some older people participated solely because their home care nurses encouraged them, while others were curious and open to the possibilities of improving their oral health or participated because of social contact. One of the main issues is that older people are seen to be ‘autonomous’ in daily oral care and they undertake oral care themselves. Older people tend to visit a dental professional only when they have a complaint of an oral problem they experience at that moment, while dental hygienists express the need for dental care that they have seen in community-dwelling older people. People with dentures tend to only visit the prosthodontist, which was not part of the OCP. The experienced impact of the education of OCP was positive and should be repeated annually, to keep home care nurses motivated, committed and aware to oral care. Collaboration between the dental hygienists and the home care nursing teams was a positive experience, but differed greatly in the participating home care nursing teams.

### 4.2. Implementations in Home Care Nursing

In line with other studies, although these studies were not completely similar to our study, the evaluation of the implementation of an oral care program uncovered the ongoing requirement of education to maintain awareness about oral health for the nursing profession [18,19,20]. The continued support of the integration of innovations in community nursing with training and education is mentioned as a key facilitator for implementation in community nursing [17]. In the implementation of OCP, we as the research team were in close contact with the home care nursing teams, and this was reported as a facilitator for implementation and resulted in a positive experience for the participants [17].

### 4.3. Oral Care and Oral Health of Community Dwelling Older People

In this study, we found that oral care behavioral change in community-dwelling older people is difficult, and this was also confirmed in another study in community-dwelling older people [23]. The importance of keeping attention on the topic of ‘oral care’ in the community nursing teams and among home care nurses has also been addressed by other researchers [24] where oral health-related issues are the ‘most missed nursing care’. In our study, older people and their home care nurses mentioned that oral care is performed by older people themselves and that autonomy is a core value in home care nursing. The concepts ‘autonomy and performing oral care yourself’ are also explained in another study, as ‘older people do oral care by themselves as long as possible’ [25]. That older people visit dental professionals for regular dental checkups less frequently is known from the literature [26], but our study showed that this could lead back to the unfamiliarity with and availability of mobile dental care (at home), and that they only visit a dental professional in case of oral problems and also older people have direct access to prosthodontists.

### 4.4. Collaboration of (Dental) Care Workers in the Community

In this study, we discovered that an introduction of the collaboration between dental hygienists and home care nurses lead to a positive experience from both sides, but that the collaboration differed greatly between the teams. Interprofessional collaboration and the integration of dental and nursing care is not further established in practice yet, as was also concluded in a recent study about the experiences with oral instruction cards in home care nursing [27]. The authors mention that ‘oral health and oral care have not yet found their place in the nursing care context’ [27] and this exactly reflects the present situation in Dutch home care nursing, despite the quite successful implementation and positive impact of OCP.

### 4.5. Costs

In OCP, a personalized oral care plan for older people which was based on an oral screening during a home visit of a dental hygienist was free of charge and provided free daily oral care products. However, the dental hygienists in our study did not feel rewarded enough for their work. To support the oral health of older people and establish access to dental care for frail groups, dental home visits by dental care professionals should be financed, possibly from public sources, whether municipal funds or perhaps included in health insurance, as suggested in a Dutch report [28].

### 4.6. Limitations and Strengths of the Study

A limitation of this study is that for some of the participants, OCP interventions and implementations in their team could have been some time ago, and this may have caused recall bias. One of the older participants was rather deaf and therefore the interview was difficult to perform.

We do acknowledge the fact that only a small sample of the original participants of the implementation of OCP was interviewed and therefore the results are possibly not transferable to the larger population involved in OCP; however, the interviewed stakeholders showed similar experiences and opinions about OCP and the implementation.

Data saturation was reached after partial analysis and the results of the qualitative evaluation are in line with the quantitative results; OCP was received positively in general.

Another limitation could be the area of implementation of OCP in the first place; OCP was implemented in one of the Northern provinces in the Netherlands, and this could have limited the generalization of the results; however, formal home care nursing is similar nationwide and therefore the results are not expected to be different across the Netherlands. A strength of this study is that the interviewers were not involved in OCP and they thus had no conflicts of interest. Participants could be open and honest in sharing their experiences without any consequences.

## 5. Conclusions and Recommendations

Future implementations of OCP should probably focus more on (older) people who do not regularly visit a dental care professional, and there should not be an age inclusion requirement of 70 years, because all people who make use of formal home care nursing are frail in some way. It is recommended that home visits and personalized oral care plans be included, since that intervention positively impacted the older participants. To enhance awareness and improve knowledge in older people, magazines or other intervention materials should be developed with older people in order to make these interventions more suitable for the target population.

Collaboration and the embedding of collaboration between home care nursing teams, dental care professionals in the community, and general community health care should be further established and encouraged by policymakers in order to provide access to dental care, even when older people themselves see no need to seek dental care. Furthermore, future implementations should also include collaboration with dental prosthodontists in order to provide denture care (at home). In future implementations, the collaboration and exchange of personalized oral care plans for older people should be structured and documented more in order to involve the home care nurses to a greater degree in the daily oral care of older people.

Collaboration and dental home visits involve costs, and this could be a barrier to future developments in oral care for community-dwelling older people. Nonetheless, this should not be the reason for dental professionals in the community to not reach out to older people and their caretakers. In addition, we encourage the Dutch government and health insurance companies to fund dental care for the Dutch population in general in the future in order to ensure access to dental care and the prevention of oral care problems that come with ageing.

## Figures and Tables

**Figure 1 ijerph-20-02124-f001:**
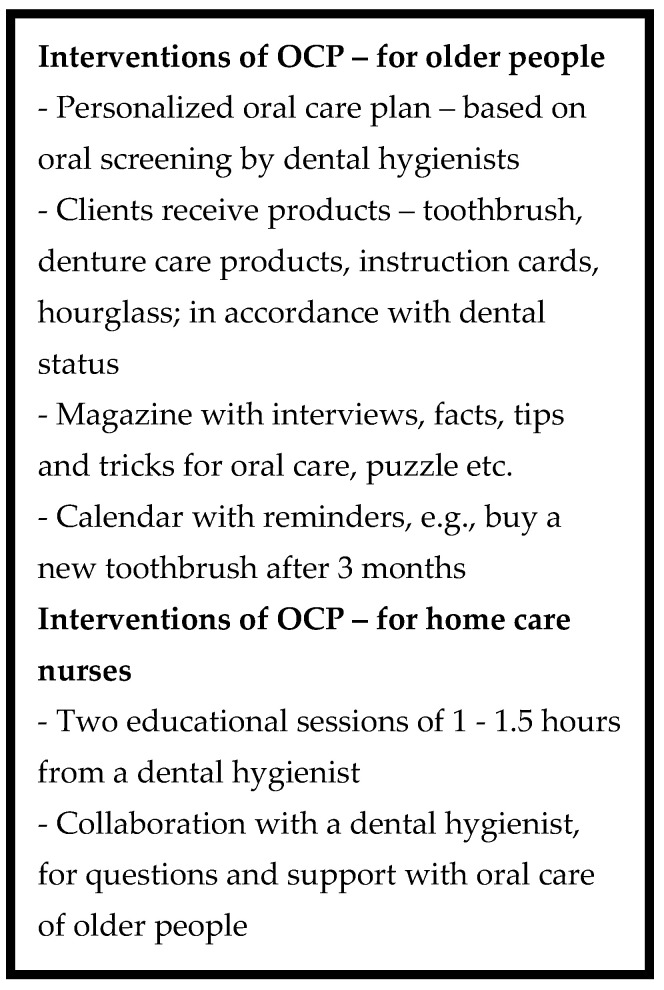
Interventions of OCP for older people and home care nurses.

**Table 1 ijerph-20-02124-t001:** Overview of codes and themes, structured by stakeholder groups.

	Home Care Nurses	Dental Hygienists	Older People
Participation in OCP	**age limitation** **costs for teams to implement OCP** **costs for clients** *clients participate because of the social aspect* * **positive experience OCP** *	**age limitation** **financial reward is limited** OCP effects on oral health (no need for second home visit) *clients participate because of social aspect* ***positive experience OCP***	participation in OCP *clients participate because of social aspect* ***positive experience OCP***
Experiences with and impact of interventions of OCP	calendar hourglass time investment ***instruction cards*** ***magazine*** ***interventions of OCP*** ***personalized oral care plan***	denture case ***instruction cards*** color/layout of OCP materials time investment ***magazine*** ***interventions of OCP*** ***personalized oral care plan***	calendar hourglass denture case ***instruction cards*** color/layout of OCP materials ***magazine*** ***interventions of OCP*** ***personalized oral care plan***
Oral health of older people	** *general health is compromised* ** *clients motivated by OCP to go to dental profs* *natural teeth/dentures* * **no complaints, no reason to see dental professionals** * *oral health care of community dwelling older people*	demand oral care seems low, need is obvious ***general health is compromised*** *clients motivated by OCP to go to dental profs* *natural teeth/dentures* ***no complaints, no reason to see dental professionals*** OCP effects oral health, second visit not needed home care nurse does not know dental status *oral health care of community dwelling older people*	***general health is compromised*** implants hurt *clients motivated by OCP to go to dental profs* *natural teeth/ dentures* ***no complaints, no reason to see dental professionals*** *oral health care of community dwelling older people*
Oral care and behavior	***autonomy/older people being independent*** **clients do oral care themselves** denture care do not visit dental profs natural teeth/dentures oral care/ADL care ***oral care behavior change*** ***oral care habits*** ***personalized oral care***	***autonomy/older people being independent*** do not visit dental profs natural teeth/dentures ***oral care behavior change*** ***oral care habits*** oral care level is higher than in nursing home ***-personalized oral care***	***autonomy/older people being independent*** **clients do oral care themselves** denture care do not visit dental profs natural teeth/dentures oral care/ADL care ***oral care behavior change*** ***oral care habits*** ***personalized oral care***
Knowledge and education	** *education/knowledge* ** **embedding of oral care and education**	** *education/knowledge* **	** *education/knowledge* **
Awareness	** *awareness* **	** *awareness* **	** *awareness* **
Communication and collaboration	**communication and collaboration** home care nurses workload **oral care/ADL care** responsibility for oral care	**communication and collaboration** home care nurse does not know dental status home care nurses workload responsibility for oral care	**oral care/ADL care**
Access and barriers to dental care	**access/barriers to dental care** *motivated by OCP to go to dental profs* ***no complaints, no reason to see dental professionals*** older people with dentures don’t see dentist *older people with dentures go to prosthetist*	**access/barriers to dental care** demand oral care seems low, need is obvious *motivated by OCP to go to dental profs* ***no complaints, no reason to see dental professionals*** not able to visit dentist because of…illness, mobility older people with dentures don’t see dentist *older people with dentures go to prosthetist*	**access/barriers to dental care** *motivated by OCP to go to dental profs* * **no complaints, no reason to see dental professionals** * *older people with dentures go to prosthetist*
Follow up of OCP	**implementation of follow up OCP** responsibility for oral care	**implementation of follow up OCP**	

Legend. Codes written in italic are mentioned by all stakeholders involved; codes written bold are mentioned more often and are major codes.

## Data Availability

The data presented in this study and interview guides (in Dutch) are available on request from the corresponding author. The data are not publicly available due to privacy or ethical reasons.

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
