# Peer review of "A Qualitative Evaluation of the Implementation of an Oral Care Program in Home Care Nursing"

_ijerph, 2023, doi:10.3390/ijerph20032124_

Round 1
Reviewer 1 Report
This article provides better access to dental care for community dwelling older people, and confirms a long term collaboration between home care nursing teams and dental care professionals in their working area should be established for the community dwelling older people.
No reversion is needed for publishment.
Reviewer 3 Report
The experiences of older people, home-care nurses and dental hygienist with the implementation of a Oral Care Program (OCP) were assessed. These experiences were overall positive, but a very selective cohort was evaluated. E.g., only elderly who wanted to participate were questioned (this were a few, it will be elderly with a special interest in OCP) and the dental hygienist who were asked were very involved with the implementation of OCP. Furthermore, no information is given concerning the background of the interviewees, therefore the data cannot be generalized of be rated of the true effects of the OCP. At best, it can be written that the very preliminary results of the implementation of a OCP program are reported. There is probably a huge bias in the experiences which cannot be truly rated as nothing is known about the age, nursing team and working area etc. It can be just a small area in the Netherlands and e.g. from just or two nursing homes. This detailed information is needed before this paper can be considered for publication.
Reviewer 4 Report
STUDY DESIGN
“A qualitative study using Thematic Analysis of semi-structured interviews was conducted after the quantitative measurements were completed (Fall 2019- Spring 2020)”
please indicate the time slot more precisely.
DATA COLLECTION
Specify that the study was carried out in the Netherlands (it is indicated in the title but it is not but then it is almost taken for granted in the text)
CONCLUSIONS
one could argue that these programs will have a much greater impact if they are open to young citizens
Reviewer 5 Report
Comments are in the pdf

Round 2
Reviewer 3 Report
The paper has improved, but a very selected sample has been studied. Only elderly from in a very restrict area (Friesland does not represent the north of the Netherlands) were asked and than only the limited group who is participating in OCP. This should be clear from the title and should be mentioned in the introduction and the aim as well as discussed (at least it should be mentioned in the limitations of this study).
Author Response
thank you for your comments and review. we have adjusted the manuscript and hereby included a response to your request.

Reviewer 5 Report
Revisions are to my satisfaction
Author Response
thank you for your review and reply.